# Phosphite Reduces the Predation Impact of *Poterioochromonas* *malhamensis* on Cyanobacterial Culture

**DOI:** 10.3390/plants10071361

**Published:** 2021-07-02

**Authors:** Narumi Toda, Hiroki Murakami, Akihiro Kanbara, Akio Kuroda, Ryuichi Hirota

**Affiliations:** Unit of Biotechnology, Division of Biological and Life Sciences, Graduate School of Integrated Sciences for Life, Hiroshima University, Hiroshima 739-8528, Japan; d215809@hiroshima-u.ac.jp (N.T.); hmuraka@hiroshima-u.ac.jp (H.M.); a-kambara@maruzenpcy.co.jp (A.K.); akuroda@hiroshima-u.ac.jp (A.K.)

**Keywords:** microalgal cultivation, cyanobacteria, contamination, predator, Chrysophyceae, *Poterioochromonas*, *Paraphysomonas*, phosphite

## Abstract

Contamination by the predatory zooplankton *Poterioochromonas malhamensis* is one of the major threats that causes catastrophic damage to commercial-scale microalgal cultivation. However, knowledge of how to manage predator contamination is limited. Previously, we established a phosphite (Pt)-based culture system by engineering *Synechococcus elongatus*, which exerted a competitive growth advantage against microbial contaminants that compete with phosphate source. Here, we examined whether Pt is effective in suppressing predator-type contamination. Co-culture experiment of *Synechococcus* with isolated *P. malhamensis* revealed that, although an addition of Pt at low concentrations up to 2.0 mM was not effective, increased dosage of Pt (~20 mM) resulted in the reduced grazing impact of *P. malhamensis*. By using unsterilized raw environmental water collected from rivers or ponds, we found that the suppression effect of Pt was dependent on the type of environmental water used. Eukaryotic microbial community analysis of the cultures using environmental water samples revealed that *Paraphysomonas*, a colorless Chrysophyceae, emerged and dominated under high-Pt conditions, suggesting that *Paraphysomonas* is insensitive to Pt compared to *P. malhamensis*. These findings may provide a clue for developing a strategy to reduce the impact of grazer contamination in commercial-scale microalgal cultivation.

## 1. Introduction

Photosynthetic microalgae have drawn great attention due to their ability to produce lipids, proteins, and other value-added chemicals from CO_2_ [1]. Their commercial application is expected to sequester 513 t of atmospheric CO_2_ and to produce more than 120 t of dry biomass per hectare annually [2]. However, despite the potential as a sustainable production tool, their commercial application remains a big challenge due to unfavorable process economics [3]. One of the major obstacles in commercial cultivation of microalgae is microbial contamination, which significantly reduces the process productivity. In microalgal cultivation, two main types of contamination mechanisms are known: nutrient competition by competitor microorganisms and predation by zooplankton that directly ingest microalgal cells [4]. Among these, the predator-type contamination often results in catastrophic damages called “pond crush”, in which predator zooplanktons quickly outcompete microalgae-preyed cells [5,6,7]. In a recent review that summarizes the feasibility of commercial-scale algal cultivation, most contamination cases were due to predator infestation, where the clearance of prey occurs within 2 to 5 days after the emergence of predators [4]. Microalgal cultivation at a commercial scale is susceptible to pest infections even in closed systems because of insufficient sterilization of culture equipment, supplemented air, and large quantities of water. As for the outdoor open-pond culture system, it is essentially inevitable to avoid contamination. Therefore, only limited microalgae strains that can grow under harsh conditions, such as high salinity (*Dunaliella*) and high pH conditions (*Spirulina*), have been demonstrated as monocultures in an open-pond culture system [3,5].

Predation, also referred to as “grazing”, is one of the trophic modes of many protozoan groups, including rotifers, ciliates, amoebas, and flagellates. In natural environments, predation plays a role in creating a selective force for the taxonomic composition of microbial communities [8]. However, once these grazers contaminate the microalgal culture, a significant reduction in microalgal biomass occurs, and an extensive decontamination process is required, causing significant damage to facilities and delayed production processes [4,9]. Recent case studies on microalgal cultivation indicate that some grazers have a limited prey spectrum [4]. However, *Poterioochromonas*, a genus belonging to the class Chrysophyceae (golden algae), indiscriminately grazes microalgae species, such as *Chlorella*, *Scenedesmus*, *Nannochloropsis*, *Dictyosphaerium*, *Chlamydomonas*, *Phormidium* [10,11], *Microcystis* [12], and *Synechocystis* [13]. Since *Poterioochromonas* rapidly grazes a large number of prey cells and multiplies within a short time, it has been recognized in recent years as a major culprit of the catastrophic pond crash of microalgal culture [4,9]. Several approaches have been developed to control *Poterioochromonas* contamination in microalgal culture by supplementation with high concentrations of CO_2_ [10], increase of pH (11.0) [13], and employment of a resistant strain [14]. However, they still have limited practical applications due to their high cost and limited host ranges for application. In addition, the life cycle, trophic mode, and physiological properties of *Poterioochromonas* remain unclear, limiting the development of effective and more feasible countermeasures.

Phosphite (HPO_3_^2−^, Pt) is an inorganic reduced P compound that is not metabolizable by general organisms. While all eukaryotes are unable to use Pt, several bacteria have been known to assimilate it [15]. Pt dehydrogenase (PtxD) is an enzyme found in Pt-assimilating bacteria that catalyzes the direct oxidation of Pt to phosphate (H_2_PO_4_^−^, Pi) [16]. Thus, heterologous expression of PtxD can confer the ability to oxidize Pt on host cells, enabling them to exert a competitive growth advantage in Pt-based culture media. This concept has been an effective selection method for prokaryotic and eukaryotic microbial hosts that do not use antibiotics [17,18]. In particular, the cheap availability of Pt offers distinct practical benefits: an industrial fermentation process without the requirement of sterilization of nutrient broth [19,20,21] and economically viable selective microbial cultivation at large scales [21,22]. Previously, we and others reported the implementation of a Pt-based selective cultivation method for cyanobacteria and several other microalgae [23,24,25]. Although this method can successfully suppress the growth of other contaminants that compete with P sources, so far no study has validated the effectiveness of Pt in suppressing the contamination by predatory zooplankton, which is the major cause of contamination observed in microalgal cultivation.

In this study, we confirmed emergence of *P. malhamensis* as a dominant predator of *Synechococcus elongatus* PCC 7942 (*Syn* 7942) from environmental freshwater in a laboratory-scale experiment. We then investigated the effect of Pt on predation of *P. malhamensis* and showed that the increased dose of Pt in *Syn* 7942 culture could mitigate the predation impact of *P. malhamensis*. Furthermore, we also examined whether the increased dose of Pt was effective in suppressing predation by other zooplankton present in the environment. Although distinct types of predators, such as *Paraphysomonas* sp., can emerge in the culture with increased Pt levels, Pt supplementation can provide a solution to address *P. malhamensis* contamination.

## 2. Results and Discussion

### 2.1. Detection of Chrysophyceae from Environmental Freshwater

To check whether predatory protists inhabiting environmental sample affect growth performance of cyanobacteria when they were co-cultured, 25 environmental water samples were collected from 10 different locations in the local rivers and ponds at Higashi-Hiroshima, Hiroshima, Japan. *Syn* 7942 was cultured using BG11-Pi containing an environmental water sample, and *Syn* 7942 growth was monitored. The emergence of Chrysophyceae (golden algae) in microalgal culture is known to cause changes in the color of the culture from green to brown or yellow due to the pigment of Chrysophyceae that predates and outcompetes microalgal cells [11,14]. Based on macroscopic observations using a time-lapse camera, *Syn* 7942 showed limited growth and color change in almost all cultures in which the clearance times were typically 1–5 days, suggesting that *Syn* 7942 cells were ingested by indigenous protists residing in the used environmental freshwater samples (Figure 1A, Appendix A). Microscopic observation found several species of predaceous planktonic organisms (e.g., amoeba and ciliate) in these cultures. We frequently observed protozoan flagellates that are single-celled with a 5–15 μm diameter, spherical or ellipsoidal, and the cells have two unequal flagella and yellow-brown chloroplasts (Figure 1B(a,b)). Fluorescence microscopic detection of chlorophyll autofluorescence revealed that the flagellates actually took up *Syn* 7942 cells inside their bodies (Figure 1B(c,d)). These morphological characteristics strongly suggest that the flagellates are highly likely to be Chrysophyceae (Ochromonadales). The frequent detection of these flagellates from freshwater environments indicated the wide distribution of Chrysophyceae in environmental water sources and strong impact on cyanobacterial cultivation.

### 2.2. Isolation, Identification, and Characterization of Chrysophyceae Emerged in Syn 7942 Cultures

To know exact species of the predatory protists, we isolated single clonal cells of Chrysophyceae from two environmental water samples: the Science Park Pond and a small artificial biotope in which Chrysophyceae Ochromonadale extensively emerged (Appendix A). Using a limiting-dilution method, two types of Chrysophyceae were obtained from each water sample and their 18S rDNA sequences (approximately 1.7 kb) were determined. As a result, one showed 100% 18S rDNA sequence identity with *Poterioochromonas malhamensis* SAG 933-1 (Accession No. EF165114), and the other showed 98% identity with that of *Ochromonas* sp. 99X4 (Accession No. MH536656.1), both of which belong to the order Ochromonadales (Figure 2). We thus designated the former strain as *P. malhamensis* TN25.3, and the latter as *Ochromonas* sp. TN25.1. Interestingly, although the sampling locations were distant and the origin of water sources of these two ponds were different, the 18S rDNA sequences of isolated *P. malhamensis* TN25.3 were completely identical to that of *P. malhamensis* SAG 933-1. Despite the large diversity of Chrysophyceae lineage [26], *P. malhamensis* has been frequently reported as a contaminant of microalgal culture across countries [9], including in this study. These facts confirm the wide distribution of *P. malhamensis* and strengthen the importance of contamination management of *P. malhamensis* in the microalgal cultivation.

### 2.3. Pt Reduces Predation Impacts of P. malhamensis on Syn 7942 Cultivation

Our previous work showed that the use of Pt on the Pt-metabolizable *Syn* 7942 culture could confer a competitive growth advantage to *Syn* 7942 and was effective in suppressing contaminants that compete with Pi in the medium [23]. However, this principle would not be applicable in suppressing grazer contamination because, for grazers, regardless of the P nutrient availability in growth media, Pi will be available by digesting prey cells. Meanwhile, Pt suppresses the growth of eukaryotic microorganisms, such as fission yeast [18] and oomycetes, a pathogenic fungus in plants [27]. Therefore, we performed a co-culture experiment of Pt-metabolizable *Syn* 7942 cells with the isolated *P. malhamensis* to determine the effect of Pt on the grazing behavior of *P. malhamensis*. When BG11-Pi was used as a medium, the clearance times were almost zero at all Pi concentrations, suggesting that *Syn* 7942 was immediately predated regardless of Pi concentration (Figure 3A). When Pt in the concentration of 0.2 mM or 2.0 mM was used, *P. malhamensis* also grazed *Syn* 7942 cells (Figure 3A), suggesting that Pt could not suppress *P. malhamensis* by means of P nutrient competition. However, the increased Pt concentration (10 and 20 mM) significantly extended the clearance time, and suppressed predation at 20 mM of Pt in the two out of four experimental batches (Figure 3A). To exclude the possibility that Pt-metabolizable *Syn* 7942 exhibited an anti-zooplanktonic effect on *P. malhamensis*, we performed a feeding experiment in which heat-killed *Syn* 7942 prey cells were fed to *P. malhamensis* TN25.3 with BG11 medium containing 20 mM of Pi or Pt. As a result, *P. malhamensis* growth was significantly suppressed by 20 mM of Pt but not Pi (Figure 3B). This suppression effect was also observed when we used *Ochromonas* sp. TN25.1, another isolate from the Science Park Pond under a different clade of *P. malhamensis* as a predator (Figure 3B). These results demonstrate a distinct effect of high dosage of Pt on reducing predation impact by *P. malhamensis* that would be expected as a novel approach for contamination management.

### 2.4. The Effect of Pt on Syn 7942 Culture Prepared with Environmental Freshwater

We next examined the potential of Pt to suppress predation on *Syn* 7942 by zooplankton residing in environmental water. We collected environmental water samples from seven different locations of the local rivers and ponds in Higashi-Hiroshima city (11 samples in total, Appendix A) and used them for the cultivation of *Syn* 7942. In both Pi and Pt, the clearance time of samples from five locations became slightly extended (median of the clearance time up to 3.9 days) as the P concentrations increased, whereas the difference between those at the same concentration of Pi and Pt was not significant (Figure 4A). However, when we used the water samples collected from the two locations of the Kurosegawa River (Points B and C), Pt supplementation at a concentration of 20 mM significantly extended the clearance time and resisted predation completely in three out of four of the repeated experiments (Figure 4B). In contrast, Pi supplementation at the same concentration did not extend the clearance time (Figure 4B). This suggested that the increased dose of Pt suppresses grazing by indigenous protists. However, we could not observe this effect using the sample collected at Point A of the Kurosegawa River, which is located approximately 8 km upstream of Point B. The Kurosegawa River is a peri-urban river classified as a Class B river in Japan (50.6 km long with 239 km^2^ area watershed) and is the drinking water source for a city in the southern area of the Hiroshima prefecture. The water samples were collected from rural areas without any remarkable inflow, such as industrial wastewater. Thus, the water quality of this river is considered not specific, but rather representing a typical river basin in Japan. To the best of our knowledge, there were no significant differences in the water quality in terms of pH, temperature, and chemical oxygen demand among all samples. One could consider that the differences among the samples were the geomorphological structure of sampling points that alters the flow discharge and water retention time, which affects the microbial community structures [28,29]. Therefore, we next investigated the microbial composition of the water sample obtained from the Kurosegawa River, in which the application of Pt mitigated the predation impact.

### 2.5. Eukaryotic Microbial Composition of the Culture Using Environmental Water Was Drastically Changed Depending on Applied P Sources on Syn 7942 Culture

To determine the reason for the application of 20 mM Pt that mitigated the predation impact of protists on *Syn* 7942 culture in the Kurosegawa River water sample, we performed a microbial community analysis by 18S rDNA amplicon sequencing using a high-throughput DNA sequencer. At day 4 of the culture using the Kurosegawa River sample (Point B) supplemented with 10 mM Pi and Pt, *Syn* 7942 cells were almost cleared by predation, and Chrysophyceae flagellates emerged (Figure 4B). From these samples, a total of 411,450 sequences of the 1422-1642 bp region of the 18S rDNA amplicon library, which generated 204 OTUs, were obtained. A taxonomic analysis of the amplicon library was performed, and their compositions were compared at the genus level (Figure 5). The dominant OTU groups in the 10 mM Pi culture were the unassigned genus that belongs to Ochrophyta (OTU-120) and the genus *Paraphysomonas*, which accounted for 67.2% and 19.6% of the population, respectively. Those in the 10 mM of Pt culture were the genus *Paraphysomonas* and *Nucleophaga*, which accounted for 50.5% and 37.0% of the population, respectively. Thus, OTU-120, an unassigned genus that belongs to the phylum Ochrophyta, was drastically reduced in the sample cultured with 10 mM of Pt. Since *Poterioochromonas* belongs to Ochrophyta, we assumed that the OTU-120 population in the culture of 10 mM Pi consisted mainly of *P. malhamensis*, which is a Pt-sensitive Chrysophyceae (Figure 3B). As expected, the 18S rDNA sequence analysis of the DNA extracted from the 10 mM Pi culture showed 100% identity with *P. malhamensis* SAG933-1 (Figure 2). Therefore, *P. malhamensis* was considered the dominant predatory species in the 10 mM Pi culture. In contrast, the genus *Paraphysomonas*, also known as a member of Chrysophyceae, was detected as the dominant species in 10 mM of Pt cultivation, suggesting that this population is less sensitive to Pt than *P. malhamensis*. *Paraphysomonas* is a colorless and phagotrophic Chrysophyceae widely present in freshwater, soil, and marine environments [30]. Sequencing analysis of the 18S rDNA clones revealed that seven out of eight clones showed 99% identity to *Paraphysomonas vestita* (Figure 2). Taken together, it is considered that the amount of *P. vestita* in the environmental water can affect the clearance time of the *Syn* 7942 culture containing 20 mM of Pt; a culture using water containing less *P. vestita* population is prone to be sensitive to Pt, whereas that using water containing *P. vestita* at a high population would be tolerant to Pt. Thus, the short clearance time observed in the sample other than that of the Kurosegawa River (points B and C) might indicate the high content of *P. vestita* in these environmental water samples (Figure 5). Therefore, quantitation of *P. vestita* cells in water prior to cultivation may be useful for predicting the success of *Syn* 7942 culture using 20 mM of Pt.

Intriguingly, to the best of our knowledge, there are no reports that *Paraphysomonas* was detected as a predator of currently practiced microalgal cultivation in both laboratory and practical settings. Therefore, we assume that either *Paraphysomonas* has been overlooked as a predator of microalgal cultivation, or that the conditions tested in this study were suitable to emergence of *Paraphysomonas*. Both *Paraphysomonas* and *Poterioochromonas* belong to the large Chrysophyceae group, and many physiological properties remain unelucidated. A more detailed analysis will be required to uncover their physiological characteristics and develop a more effective way to prevent damage during the practical cultivation of microalgae.

## 3. Materials and Methods

### 3.1. Bacterial Strains and Culture Conditions

*Syn* 7942 and its derivative strains used in this study were routinely cultured at 30 °C with 2% CO_2_ bubbling under continuous illumination (50 μmol photons/m^2^/s) in a modified BG11 medium containing 20 mM N-Tris(hydroxymethyl)-methyl-2-aminoethanesulfonic acid (TES)-KOH (pH 8.0) and 0.2 mM phosphate (BG11-Pi) or phosphite (BG11-Pt) as a phosphorous source [23]. These media were prepared by mixing 5 mL of the stock solutions for the BG11 medium and phosphorus (P) source, and 45 mL of sterilized Milli-Q water. Pt used in this study was a 50% potassium phosphite (K_2_HPO_3_) solution purchased from Omichi Seiyaku (Omichi Seiyaku Co., Ltd., Osaka, Japan). The Pt stock solution was diluted to 1 M with Milli-Q water, filter-sterilized, and stored at −20 °C until use. Two Pt-metabolizable *Syn* 7942 strains, AK024 and RH714, were cultured on BG11-Pt. AK024, a derivative strain of *Syn* 7942, in which *Ralstonia* sp. *ptxABCD* is integrated at the neutral site I (NSI) [23], is capable of assimilating both Pi and Pt because the strain expresses endogenous Pi transporters (PstSCAB and Pit), Pt/Pi transporters PtxABC, and Pt dehydrogenase (PtxD) for Pt uptake and oxidation. RH714 is a biologically contained *Syn* 7942 strain that can only grow on Pt but not Pi, because it is devoid of its endogenous Pi transporters and expresses a Pt-specific transporter HtxBCDE and PtxD [23]. For culturing AK024 and RH714, 40 μg/mL of spectinomycin and 0.2 mM isopropyl-β-D-thiogalactopyranoside (IPTG) were added to the culture medium. In addition, for culturing RH714, 10 μg/mL of kanamycin and 2 μg/mL of gentamycin were added to the culture medium. Although RH714 showed slightly reduced growth in BG-11(Pt) compared to AK024 [23], this difference was negligible under the conditions tested in this study.

### 3.2. Cultivation Using Environmental Freshwater Samples

The environmental freshwater samples were collected from natural rivers and ponds in Higashi-Hiroshima city, Hiroshima, Japan, throughout the years (Dec. 2017–Mar. 2021, Appendix A). The environmental freshwater samples were collected, filled in pre-cleaned 1 L polyethylene bottles, and kept at 4 °C until they were used for the experiments performed within a day after collection. The sampling locations are listed in Appendix A. The collected environmental freshwater samples were used without sterilization; however, large debris in the samples was removed by passing through a paper towel. To isolate *Poterioochromonas* from environmental freshwater, *Syn* 7942 was cultured in 50 mL of BG11-Pi or -Pt, in which a portion of sterilized Milli-Q water (45 mL) was replaced with the collected environmental freshwater samples. A 0.5 mL of fully grown *Syn* 7942 culture was inoculated in a glass test tube of 3 cm diameter and 20 cm length containing 50 mL of BG11-Pi/Pt prepared with the environmental freshwater samples to give a *Syn* 7942 cell density of approximately 10^7^ cells/mL. To determine the effect of P concentration on grazing by protists, P concentrations varying from 0.2 to 20 mM were used. The test tube was capped with an adaptor and a cotton plug, which passes 2% CO_2_/air aseptically and was cultured under the abovementioned conditions. During cultivation, macroscopic images of the test tubes were recorded using a time-lapse camera (Brinno TLC200) at 10 min intervals for 14 days to monitor cell growth and the emergence of grazers. After data collection, the recorded time-lapse video data was analyzed by Apple iMovie software (Apple Inc.) to measure the duration that *Syn* 7942 persists in the culture until completely cleared by predation, which we termed “clearance time.” The clearance time was determined by measuring the duration from the beginning of the culture until the color of *Syn* 7942 culture disappeared by predation or turned brownish due to the pigment of the golden algae. Microscopic images were captured using a CCD camera (DP74, Olympus, Tokyo, Japan) mounted on an Olympus fluorescence microscope (BX52, Olympus, Tokyo, Japan). Chlorophyll autofluorescence images were observed under fluorescent microscopy using a filter unit U-MWIG3 (Olympus; excitation: 530-550 nm, emission: 575-625 nm, dichroic mirror: 570 nm).

### 3.3. Identification of Protists in Environmental Freshwater

To identify protists that emerged in the culture, 1 mL of the culture was collected, centrifuged, and the pellet was subjected to DNA extraction using the PowerSoil DNA Extraction Kit (Qiagen). The extracted DNA was used as a template for PCR by targeting the 18S rDNA nucleotide sequence. The PCR was performed with a 20 μL reaction mixture containing 0.2 units of KAPA Taq EXtra DNA polymerase (KAPA Biosystems, South Africa), 0.5 mM dNTPs, 1.5 mM MgCl_2_, 0.5 μM primers, 10 ng of extracted DNA, and the reaction buffer supplied with the enzyme. The primers EukA (5′-AACCTGGTTGATCCTGCCAGT-3′) and EukB (5′-TGATCCTTCTGCAGGTTCACCTAC-3′) [31] were used to amplify approximately 1700 bp of 18S rDNA sequence. To amplify the 18S rDNA sequence of *Paraphysomonas* sp., 836_EukF2 (5′-TACTGTGAAACTGCGAATGG-3′) and 838_EukR3 (5′-CGGGCGGTGTGTACAAAGGG-3′) were used. The PCR conditions were as follows: 94 °C for 3 min, 30 cycles of 94 °C for 2 min, 53 °C for 50 s, and 72 °C for 30 s, followed by a final extension step at 72 °C for 5 min. The PCR products were separated by agarose gel electrophoresis, and the amplified DNA was purified by using the Monarch DNA Gel Extraction Kit (New England Biolabs, Beverly, MA, USA). The purified DNA was then cloned into the pGEM-T easy vector (Promega, Madison, WI, USA) to separate DNA amplified from different template sources. The ligated plasmids were transformed into *Escherichia coli* DH5α and selected on LB agar plates containing 50 mg/L ampicillin, 0.004% 5-bromo-4-chloro-3-indolyl-β-D-galactopyranoside (X-gal), and 0.5 mM isopropyl-β-D-thiogalactopyranoside (IPTG). After incubating the plates at 37 °C for 12 h, the positive colonies were chosen, and colony PCR was performed in a 20 μL reaction volume containing 10 μL of KOD One PCR Master Mix (TOYOBO, Tokyo, Japan), 0.5 μM of each of EukA and EukB or 836_EukF2 and 838_EukR3 primers. The PCR conditions were as follows: 98 °C for 2 min, 30 cycles of 98 °C for 10 s, 55 °C for 5 s and 68 °C for 10 s, followed by a final extension step at 68 °C for 5 min. After the PCR products were purified as described above, their nucleotide sequences were determined by Fasmac DNA sequencing service (Fasmac, Atsugi, Japan), using the primers EukA, EukB, 584f_fw (5′-TCCACCAACTAAGAACGGCC-3′), 836_EukF2, and 838_EukR3. If the *Syn* 7942 culture was dominated by a specific protist species, the amplified 18S rDNA was directly sequenced without cloning.

### 3.4. Isolation of P. malhamensis from Environmental Freshwater Samples

To isolate *P. malhamensis*, *Syn* 7942 cultures containing environmental freshwater samples were subjected to the limiting dilution method. The serially diluted culture suspension using BG11-Pi was transferred into each well of a 96-well plate at a final density of 0.1 protist cells/100 μL/96-well with *Syn* 7942 prey (approximately 10^5^ cells/well). After 8 days of incubation at 30 °C under illuminated conditions, microwells without protists showed a green color because of the growth of *Syn* 7942. Meanwhile, protist-positive microwells were recognized as having a clear color due to protist predation on *Syn* 7942. Such cultures were serially expanded in BG11-Pi with *Syn* 7942 prey cells, and a single-cell lineage was established. During the passaging culture, kanamycin (50 mg/L) and gentamycin (2 mg/L) were added to kill the contaminating bacteria. After isolation, *P. malhamensis* monoculture was confirmed by light microscopy, and the 18S rDNA sequence of the isolates was validated. Isolated Chrysophyceae strains were routinely cultured in 200 mL Erlenmeyer flasks containing 50 mL of BG11-Pi inoculated with approximately 10^6^ *Syn* 7942 cells under continuous illumination (50 μmol photons/m^2^/s) and ambient air.

### 3.5. Co-Culture Experiment of Synechococcus with Isolated P. malhamensis

For co-culture experiment, cell numbers of *Syn* 7942 and *P. malhamensis* were determined based on the value of optical density at 750 nm (OD_750_ of 1.0 corresponded to 10^8^ cells/mL) and direct cell counting using a bacteria counting chamber (Erma, Tokyo, Japan), respectively. Then, 10^8^ cells of *Syn* 7942 and 5 × 10^5^ cells of *P. malhamensis* were inoculated in a glass test tube containing 50 mL of BG11 containing 0.2–20 mM of Pi or Pt. The culture equipment, culture conditions, and data collection method were described as previously (see the section “*Cultivation using environmental freshwater samples*”). For the feeding experiment, *Syn* 7942 cells were autoclaved at 121 °C for 15 min and used as prey cells and the culture was done by using a 200 mL Erlenmeyer flask with orbital shaking under ambient air.

### 3.6. Phylogenetic Analysis

All known 18S rDNA sequences of Chrysophyceae were obtained from the GenBank database (https://www.ncbi.nlm.nih.gov/genbank/, accessed on 11 June 2021). Nucleotide sequences were aligned using MAFFT v7.475 [32] and the highly divergent regions of the alignment were further trimmed out with Gblocks v0.91b, allowing for gaps in the final alignment [33]. Approximate maximum-likelihood trees were constructed using MEGA6 based on Tamura 3-parameter +G+I model [34]. The reliability of the internal branches was assessed using the nonparametric bootstrap method with 1000 replicates.

### 3.7. Microbial Community Analysis

Microbial community analysis was performed by amplicon sequencing of the partial 18S rDNA sequence using an NGS approach on an Illumina platform. A 1 mL sample of day 4 culture of *Syn* 7942 using the environmental freshwater samples collected from the Kurosegawa River (Point B, collected on 10 October 2018) was pelleted by centrifugation (15,000× *g*, 4 °C), and total DNA was extracted as described previously. The partial sequence of eukaryotic 18S rDNA was amplified by PCR using the primer pairs 1422f (5′-adaptorA-ATAACAGGTCTGTGATGCC-3′) and 1642r (5′-adaptorB-CGGGCGGTGTGTACAAAGG-3′) [35], in which the sequences for adaptor A and B were 5′-ACACTCTTTCCCTACACGACGCTCTTCCTCCGATCT-3′ and 5′-GTGACTGGAGTTCAGACGTGTGCTCTTCCGATCT-3′, respectively. The first PCR amplification was carried out in a 20 μL reaction volume containing 1 ng template DNA, 0.5 μM of primers, 0.2 mM dNTPs, and 1 unit of Takara Ex Taq (Takara Bio, Shiga, Japan). The PCR conditions were as follows: 94 °C for 2 min, 30 cycles of 94 °C for 20 s, 50 °C for 15 s, and 72 °C for 15 s, followed by a final extension step at 72 °C for 5 min. The resulting amplification product (approximately 220 bp) was separated by agarose gel electrophoresis, purified as described above, and cleaned using AMPure XP beads (Beckman Coulter, Brea, CA, USA). The second PCR and subsequent steps of library preparation and sequencing were performed by Bioengineering Lab. Co., Ltd. (Seibutsu Giken Inc., Atsugi, Japan). Amplicons were sequenced using the MiSeq system (Illumina, Inc., San Diego, CA, USA) using 2 × 300 bp paired-end reads. The obtained DNA sequences were used for genus-level phylogenetic analysis using the QIIME 2.0.

### 3.8. Nucleotide Sequence Accession Numbers

The 18S rDNA sequence data of the isolated strains in this study are available in DDBJ under the accession numbers LC636300 to LC636306. The 18S rDNA amplicon sequence data were deposited in the DDBJ nucleotide sequence database under the accession number DRA012179 and DRA012194.

## 4. Conclusions

In this study, by using Pt-assimilating *Syn* 7942 strains, we found that the growth of *P. malhamensis* was suppressed by the supplementation of Pt in BG11 growth medium. When Pt was applied to *Syn* 7942 culture using unsterilized environmental water, *Paraphysomonas*, a Chrysophyceae that has rarely been reported as a grazer of microalgae, emerged instead of *Poterioochromonas*. This study provides a potential direction for the contamination management of notorious predatory grazers in commercial-scale microalgal cultivation.

## Figures and Tables

**Figure 1 plants-10-01361-f001:**
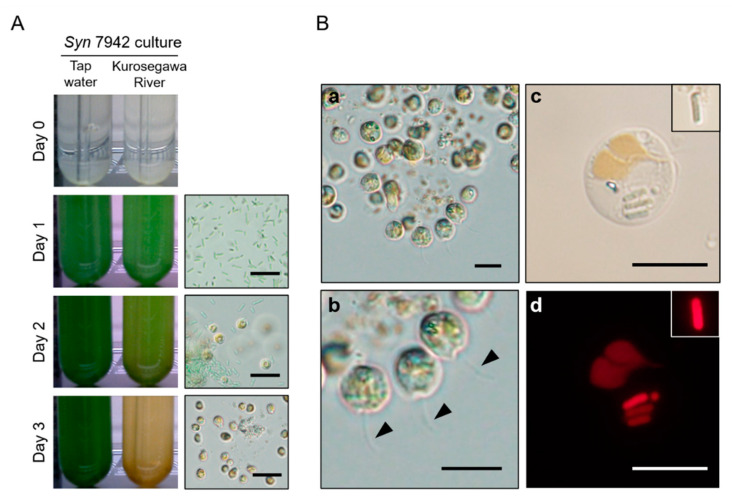
Detection of Chrysophyceae in *Syn* 7942 culture. (**A**) Macroscopic pictures of the typical *Syn* 7942 culture on BG11-Pi prepared with unsterilized environmental freshwater (left pictures), and their corresponding microscopic pictures (right pictures). The experimental result using a Kurosegawa River water sample (Appendix A. Sample 12) was shown. Tap water was used for the control experiment. Microscopic pictures are only shown for the culture using environmental water. Scale bars, 20 μm. (**B**) Detailed microscopic pictures of Chrysophyceae detected in the culture prepared with the environmental water sample (**a**–**d**). A colonial structure of the detected Chrysophyceae (**a**). The arrowheads show one of the flagella, whereas the other is indistinguishable in the photographic display (**b**). Representative image of Chrysophyceae digesting *Syn* 7942 cells (**c**), and corresponding autofluorescence image of *Syn* 7942 (**d**). The top right insets show a cyanobacterial cell free from grazing in the same culture (**c**,**d**). Scale bars, 10 μm.

**Figure 2 plants-10-01361-f002:**
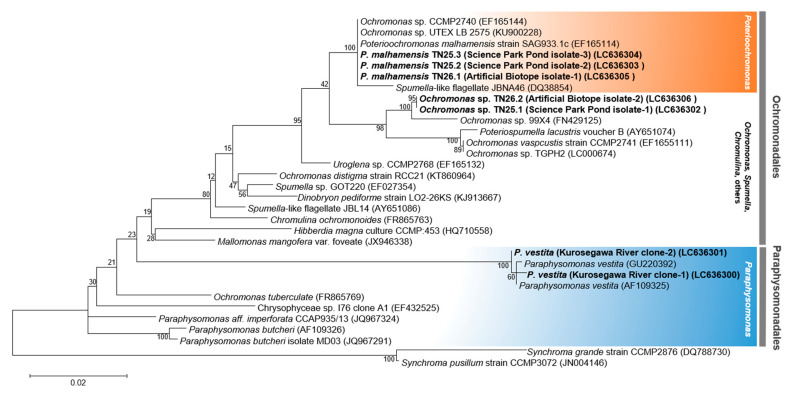
Maximum likelihood tree of the 18S rDNA of Chrysophyceae and the isolated strains in this study. A phylogeny of Chrysophyceae including strains isolated in this study was constructed by maximum likelihood method based on 18S rDNA sequence. *Synchroma grande* CCMP2876 and *Synchroma pusillum* CCMP3072 were used as outgroups. The isolated strains in this study are shown in bold. *Poterioochromonas* and *Paraphysomonas* are highlighted. Bootstrap values are indicated at each branch. Scale bar, 0.02 substitutions per site.

**Figure 3 plants-10-01361-f003:**
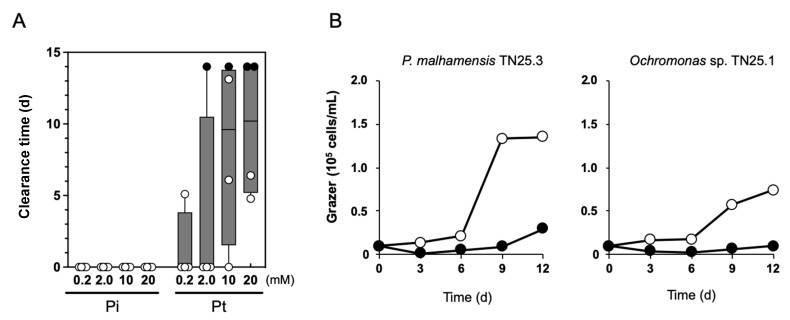
Effect of different concentrations of Pi and Pt on *P. malhamensis* in *Syn* 7942 culture. (**A**) The clearance time of co-culture experiment of *Syn* 7942 with *P. malhamensis* is represented by a box-and-whisker plot. Four independent experimental results are shown in plots. The filled plots represent the *Syn* 7942 culture that resisted predation until the culture day 14, the end of the cultivation. For the cultivation of BG11-Pi and -Pt, *Syn* 7942 and RH714, a Pt-metabolizable *Syn* 7942 strain (see “Materials and Methods” for more details), were used, respectively. (**B**) The growth of *P. malhamensis* TN25.3 and *Ochromonas* sp. TN25.1 under feeding of heat-killed *Syn* 7942 with different P sources. Open plots, Pi. Filled plots, Pt. The data are representative of two independent experiments with essentially the same results.

**Figure 4 plants-10-01361-f004:**
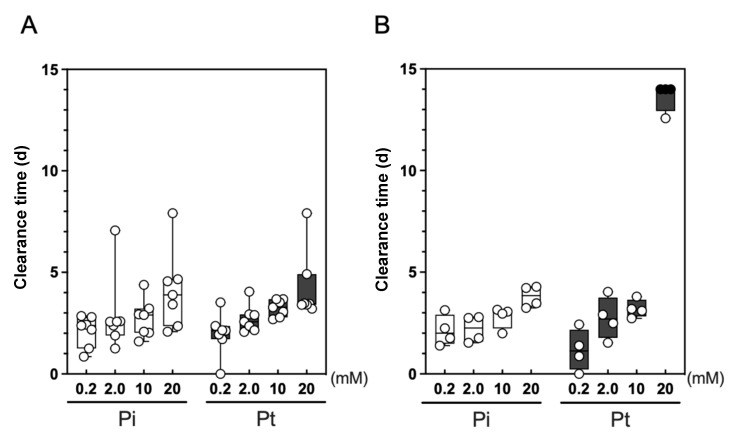
Effect of different concentrations of Pi or Pt on grazers inhabiting freshwater environments. The clearance times of *Syn* 7942 culture using unsterilized environmental water samples collected from five locations including rivers and ponds (**A**), or the Kurosegawa River Points B and C (**B**) (Appendix A), are shown with box-and-whisker plots. The open plots indicate the clearance time of the culture, and the filled ones represent the completion of culture.

**Figure 5 plants-10-01361-f005:**
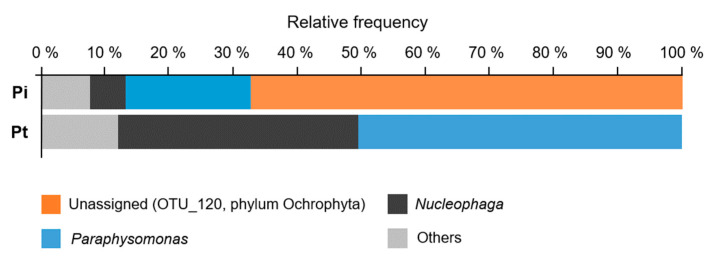
The genus-level eukaryotic microbial compositions on *Syn* 7942 cultures using the environmental water from the Kurosegawa River. The bar chart shows the relative abundance of major eukaryotic taxa in the *Syn* 7942 culture prepared with the environmental raw water collected from the Kurosegawa River (Point B). The cultures were supplemented with either 10 mM Pi or Pt, and the samples were collected at day 4 (Figure 4B, see Materials and Methods).

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
