# Peer review of "Phosphite Reduces the Predation Impact of Poterioochromonas malhamensis on Cyanobacterial Culture"

_plants, 2021, doi:10.3390/plants10071361_

Round 1

Reviewer 1 Report

Poterioochromonas is one of the main harmful grazer in the commercial microalgae culture. Effective control methods are still insufficient. This study provided a novel method (i.e., phosphite) with possibility to control the Poterioochromonas. It’s very interesting. I have few minor queries to be addressed before accepting for publication.

  1. On the identification of Poterioochromonas, lorica is a very important feature to distinguish Poterioochromonasand its sister genus Ochromonas. Have you checked if the P. malhamensis isolated in this study could form the structure of lorica. You can stain the larica with Calcofluor White as references (Herth,1980, Journal of cell biology, 87, 442-450; Peck, 2010, Protist, 161,148-159).  

  1. The yellow-brown chloroplast in Poterioochromonas(Fig. 1c) should also be red in autofluorescence image (Fig. 1d).

  1. 3B and Fig. 3C, 1) error bars should be added; 2) number of biological duplicationshould also be exhibited in the end of the figure legend. For sample, n=3; 3) title of Y-axis could be expressed as ‘Grazer (105 cells/mL)’, then the numerical value of Y-axis can be simple 0.5, 1,1.5,2…

Reviewer 2 Report

The work by Toda et al., entitled “Phosphite reduces predation impact of Poterioochromonas malhamensis on cyanobacterial culture” and submitted to the journal Plants, presents a set of observations that indicate that the well-described, predatory, cyanobacterial and microalgal grazer P. malhamensis, isolated from field samples, is sensitive to the presence of phosphite (supplied as K2HPO3), while phosphite-assimilating Synechococcus elongatus PCC 7942 thrive in such conditions. In addition, the work also shows that, using environmental water samples, a different genus of golden algae emerges in phosphite supplemented growth media (Paraphysomonas). This work is relevant for those interested in cultivating cyanobacterial strains in outdoor settings, which is essential towards commercial-scale cyanobacterial (and possibly algal) cultivation.

The manuscript is very well written, the message is clear and straightforward, and the results are sound. In addition, the experimental procedures described seem fit for the questions raised throughout the work.

Nevertheless, a few aspects require the authors’ attention, namely:

Line 17: was dependent, and not “was depended”

Line 33: remove “a” in “is a microbial contamination”

Lines 34-35: remove “to be emerged”

Lines 37-38: microalgae preyed cells, and not “microalgae prey cells “

Line 67: “Phosphite (H3PO3, Pt)”: Phosphite is an anion, and H3PO3 is phosphorous acid. Please modify accordingly.

Lines 77-78: “Previously, we reported…”. As the references supporting this sentence are not exclusively the work of the authors of this manuscript, it is recommended to include “we and others reported”.

Lines 125-126: “The arrowheads show one of flagella, whereas another is indistinguishable in the photographic display.”

It should be “The arrowheads show one of the flagella, whereas the other is indistinguishable in the photographic display.”

Details should be provided as to the settings used for acquisition of the fluorescence micrographs (or is it confocal microscopy)? Excitation light wavelength, emission light wavelength, etc. are important to include. This is especially relevant because the authors refer to the fact that chlorophyll autofluorescence was investigated (line 110) and those settings are therefore required to assess whether other pigments’ fluorescence was equally observed.

Line 188: “Syn 7942 and RH714 were used”

The only place in the text where the authors describe the Pt-metabolizing Syn 7942 strains used in this study is in the “Materials and Methods”. As this section is located after the section “Results and Discussion”, it is not easy to understand what RH714 is. Thus, it is recommended that the authors include a short mention in the text (under the section “Results and Discussion”) to present which Pt-metabolizing Syn 7942 strains were used and in which conditions.

Line 197: Table S2

Table S2 is broken in to 6 different pages, which makes the data very difficult to read and interpret. It is recommended that the authors present the table in a different format. Maybe possible to have it in a single page.

Lines 260-262:

“and the water containing less P. vestita population is prone to be sensitive to Pt, but the water containing P. vestita at a high population would be tolerant to Pt.”

This part of the sentence is difficult to interpret: How can the water be “sensitive” or “tolerant” to Pt? Please rephrase.

Line 378: contaminating instead of “contaminated”.

Line 439: remove “a” in “… management of a notorious predatory grazers …”
